# Proteomic and Metabolic Analysis of *Pinus halepensis* Mill. Embryonal Masses Induced under Heat Stress

**DOI:** 10.3390/ijms24087211

**Published:** 2023-04-13

**Authors:** Cátia Pereira, Ander Castander-Olarieta, Itziar A. Montalbán, Vera M. Mendes, Sandra Correia, Ana Pedrosa, Bruno Manadas, Paloma Moncaleán, Jorge Canhoto

**Affiliations:** 1Centre for Functional Ecology, TERRA Associate Laboratory, Department of Life Sciences, University of Coimbra, Calçada Martim de Freitas, 3000-456 Coimbra, Portugal; catia.pereira@student.uc.pt (C.P.); sandraimc@uc.pt (S.C.); anasimoespedrosa@gmail.com (A.P.); 2Department of Forestry Science, NEIKER-BRTA, 01192 Arkaute, Spainimontalban@neiker.eus (I.A.M.); 3CNC—Center for Neuroscience and Cell Biology, University of Coimbra, 3000-456 Coimbra, Portugal; vmendes@cnc.uc.pt (V.M.M.); bmanadas@cnc.uc.pt (B.M.); 4InnovPlantProtect CoLAb, Estrada de Gil Vaz, 7350-478 Elvas, Portugal

**Keywords:** Aleppo pine, conifers, metabolism, proteins, stress response

## Abstract

Understanding the physiological and molecular adjustments occurring during tree stress response is of great importance for forest management and breeding programs. Somatic embryogenesis has been used as a model system to analyze various processes occurring during embryo development, including stress response mechanisms. In addition, “priming” plants with heat stress during somatic embryogenesis seems to favor the acquisition of plant resilience to extreme temperature conditions. In this sense, *Pinus halepensis* somatic embryogenesis was induced under different heat stress treatments (40 °C for 4 h, 50 °C for 30 min, and 60 °C for 5 min) and its effects on the proteome and the relative concentration of soluble sugars, sugar alcohols and amino acids of the embryonal masses obtained were assessed. Heat severely affected the production of proteins, and 27 proteins related to heat stress response were identified; the majority of the proteins with increased amounts in embryonal masses induced at higher temperatures consisted of enzymes involved in the regulation of metabolism (glycolysis, the tricarboxylic acid cycle, amino acid biosynthesis and flavonoids formation), DNA binding, cell division, transcription regulation and the life-cycle of proteins. Finally, significant differences in the concentrations of sucrose and amino acids, such as glutamine, glycine and cysteine, were found.

## 1. Introduction

Forest ecosystems present an uncountable environmental and economic value; notwithstanding, breeding forest trees for abiotic stress tolerance by conservative techniques is difficult [1]. The efficiency of modern forest management and breeding programs highly depends on our understanding of the mechanisms involved in their adaptation to stress conditions.

Aleppo pine (*Pinus halepensis* Mill.) is a Mediterranean species with significant ecological plasticity, which permits it to live in poor calcareous soils where other tree species have difficulties to thrive [2]. Furthermore, it is considered a species adapted to fire and drought and, as a pioneer species, its introduction in degraded areas can facilitate the long-term colonization and expansion of late-successional species [3,4]. These characteristics make this species a potential alternative for reforestation. However, the foreseen climatic changes and the interaction between warm temperatures and precipitation can modify the success of its post-fire recovery patterns and resilience in the future [5].

Heat stress is frequently defined as a period in which temperatures are substantially high during a sufficient amount of time that causes permanent damage to functions or the correct development of plants [6]. Moreover, heat is usually accompanied by drought, and both stresses present overlapping roles [7,8,9]. Nonetheless, the stimulation of induced defense and adaptation responses by “priming” plants with stressful temperatures during both zygotic [10] and somatic embryogenesis (SE) [11,12] seems to favor plants’ resilience to extreme temperature conditions through a “memory” formation.

In this sense, micropropagation of conifers through SE has been widely studied as a promising technique for improving the effectiveness of breeding programs [13,14]. This biotechnological tool, which allows the formation of somatic embryos from somatic cells [15,16], brings the opportunity for large-scale propagation of “elite” plants [17] and is an effective model system to analyze the physiological and biochemical events occurring during embryo induction and development as well as responses to abiotic stresses [18,19,20].

In plants, all abiotic stresses induce a cascade of physiological and molecular events that may lead to similar final responses that impact metabolic pathways (alterations in the levels of crucial metabolites, proteins, or transcription factors) and/or epigenetic memory [21,22]. Metabolic memory may appear as a possible consequence of epigenetic regulations and is related to changes in metabolites, enzymes and proteins that could result in phenotypic changes [23].

Since the first results on SE in *Pinus halepensis* were achieved in our laboratory [24], SE has been used to understand better how this species behaves under abiotic stresses. Initial experiments showed that changes in temperature and water availability in the early stages of the process determine not only the success of its different stages but also the final outcome of the process [25,26]. Considering those results, extreme temperature treatments were applied during the induction of embryonal masses (EMs) to realize if a “priming” effect could be obtained. Those heat stress treatments modulated, again, the efficiency of the process itself, as well as the morphology and endogenous hormonal profiles of the resultant EMs [27]. Then, the effect of the same treatments on the global DNA methylation rates and the differential expression of stress-related genes at EMs and needles from in vitro somatic plants were evaluated: even though no significantly different levels of DNA methylations were found, small fluctuations were presumably related to the success of different stages of SE and long-lasting variations in the differential expression of stress-related genes were found [28]. The study of stress-responsive genes is crucial for the understanding of plant adaptation [29]; nevertheless, the characterization of how the expression of those genes is regulated through posttranscriptional and post-translational modifications offer a deeper level of knowledge of the molecular mechanisms governing plant adaptation and stress tolerance [30,31].

In the present work, in order to complement our understanding of the stress mechanisms triggered by temperature stress during the induction of SE in Aleppo pine, the study of the proteome of EMs produced under the high temperatures referred to above was conducted. Proteomics is a powerful tool to determine proteins’ identity, abundance and post-translational modifications in association with stress responses [32]. The accumulation of heat shock proteins (HSPs) under the control of heat shock transcription factors is known to play a central role in the heat stress response and in acquired thermotolerance in plants [33]. Moreover, the mechanism involved in the scavenging of stress, which produces reactive oxygen species (ROS) and can provoke severe cellular damage, primarily involves an enzymatic-regulated system [34]. In turn, metabolites reflect the integration of gene expression, protein interaction, as well as other different regulatory processes, and metabolomics has been widely used as a research tool to elucidate how plants respond to abiotic stress [35,36]. In this sense, we also analyzed the different concentrations of soluble sugars, sugar alcohols and amino acids between induction treatments.

## 2. Results

### 2.1. Relative Quantification of Proteins

The search performed against the 4256 protein sequences reviewed at the Viridiplantae database allowed the identification of 1315 proteins, and relative protein quantification was attained for 858 proteins. The primary application of PLS-DA multivariate analysis of these 858 proteins allowed for the reduction of the complexity of the results and for the sample groups’ visualization. As shown in Figure 1a, the differences between the four induction treatments, analyzed by the bi-dimensional representation of their score’s plots, did not completely separate the loading plots. Nevertheless, it appears that the first component of PLS-DA pairwise comparison models assembled the variability related to heat stress response; as the induction temperature applied increased, so did the corresponding values along this component. The complexity of the results obtained at the second component may be associated with the variability between samples.

At the same time, this multivariate analysis allowed for the identification of the most significant proteins able to classify the four conditions based on the variable influence on projections (VIP) values (Figure 1b). A cut-off of 1 was defined to select the important variables, considered the best classifiers for the group separation observed in the scores plot of the four conditions. In this sense, 267 proteins were selected and considered the ones to be involved in heat stress responses.

As a preliminary step to interpreting the data and categorizing these proteins, a gene ontology (GO) enrichment analysis was performed using FunRich and the Plants database from the UniProt database. It allowed us to extract information about the cellular components with which these 267 proteins are associated (Figure 2a), as well as the molecular functions (Figure 2b) and the biological processes (Figure 2c) that they are involved in. Cytosol (32.5%), cytoplasm (26.4%) and mitochondrial (15.5%) proteins were the ones with the highest representation concerning cellular location (Figure 2a). Regarding their molecular function, two major functional groups appeared: proteins involved in metabolism (ATP binding, endopeptidase activity, threonine-type endopeptidase activity, phosphoglocunate (decarboxylating) activity and fructose-bisphosphate aldolase activity) that account for 30.2%, and signaling-involved proteins (GTP binding, GTPase activity and mRNA binding) that account for 18% (Figure 2b). Finally, concerning the biological processes, response to cadmium ion (7.2%), proteasome-mediated ubiquitin-dependent protein catabolic process (4.2%), the tricarboxylic acid cycle (3.8%) and the glycolytic process (3.8%) presented the highest representativeness.

The parallel univariate analysis performed with the Kruskal–Wallis test identified 34 proteins with significant differences (*p* < 0.05) between the four induction treatments (Appendix A). These results, in combination with the ones previously obtained by PLS-DA analysis, allowed the identification of the 27 proteins with higher interest concerning heat stress response (Figure 3). The detailed list of these proteins, including their accession number, identified species, the fold-change obtained between conditions and the results from both statistical analyses, is presented in Table 1.

Finally, a hierarchical clustering heatmap (Figure 4) was generated to better visualize the differences between treatments regarding the top 27 proteins’ relative accumulation. Two main groups were clearly defined. The first main group corresponds to proteins with decreased amounts at samples induced under higher temperatures relative to the control (condition 1). Within this main group, different patterns were found (Table 1, Figure 4). Thus, polyadenylate-binding protein RBP47 (a protein involved in RNA binding and its aggregation in the cytoplasm) and aconitate hydratase (involved in the biosynthesis of carbohydrates) presented significantly decreased amounts in samples from 40 °C (4 h) (condition 2) and 50 °C (30 min) (condition 3). On the other hand, it was found that a 25.3 kDa vesicle transport protein, protein kinase G11A, and molybdenum cofactor sulfurase, three proteins with catalytic activity enrolled in metabolism regulation and signal transduction processes, had significantly decreased amounts in samples from 50 °C (30 min) and 60 °C (5 min) (condition 4). It should also be noted that the amounts of 20 kDa chaperonin, probable fructokinase-7, and agglutinin were decreased at all heat stress treatments applied.

The second main group gathers the proteins with increased amounts in samples from heat stress treatments. Apart from the heat-shock protein chaperonin CPN60, which presented the highest fold change between 40 °C (4 h) and the control (Table 1), proteins presented higher concentrations at 50 °C (30 min) and 60 °C (5 min). It seems important to mention that the majority of the proteins with increased amounts at samples induced at higher temperatures include enzymes directly involved in the regulation of metabolism: a high number of proteins with increased concentrations under heat stress conditions implicated glycolysis and the tricarboxylic acid cycle (pyruvate dehydrogenase E1 component subunit beta-4, citrate synthase, pyruvate dehydrogenase E1 component subunit beta-1, trifunctional UDP-glucose 4.6-dehydratase/UDP-4-keto-6-deoxy-D-glucose 3.5-epimerase/UDP-4-keto-L-rhamnose-reductase RHM1, 6-phosphogluconate dehydrogenase decarboxylating 1), a membrane-bound enzyme required for electron transfer from NADPH to cytochrome P450 (NADPH--cytochrome P450 reductase 1), as well as amino acid biosynthesis (3-isopropylmalate dehydrogenase, carbamoyl-phosphate synthase large chain) and flavonoids formation (chalcone synthase), were identified. Additionally, proteins involved in DNA binding and cellular division (dynamin-related protein 12A, actin-depolymerizing factor 10), transcription regulation (histone H4 variant TH091, histone H2A.2.2) and the life-cycle of proteins, such as those directly related with nuclear (40S ribosomal protein S15a-1, 60S ribosomal protein L18-3) and mitochondrial (glutamyl-tRNA (Gln) amidotransferase subunit B) translation, were present in this group. NADP-dependent malic enzyme 3 and alcohol dehydrogenase presented significantly lower amounts at 60 °C (5 min) compared to 40 °C (4 h).

### 2.2. Relative Expression of Stress-Related Transcripts

Nine top proteins were selected, and the primers for their respective transcripts were designed. Afterwards, the relative expression patterns of the selected stress-related transcripts with respect to the control (23 °C) for the different induction temperature treatments were quantified using an RT-qPCR approach.

The stress-related transcripts analyzed did not present statistically significant differences between treatments (Figure 5, Appendix A). Nonetheless, *20 kDa CHAPERONIN (CPN20)*, *CHAPERONIN CPN60 (CPN60)*, *CITRATE SYNTHASE (PDHB)*, *PYRUVATE DEHYDROGENASE E1 COMPONENT SUBUNIT BETA-1* and *6-PHOSPHOGLUCONATE DEHYDROGENASE, DECARBOXYLATING 1 (G6PGH1)* presented a similar expression pattern: a lower expression in samples from 40 °C (4 h), a higher expression in samples from 60 °C (5 min) and a similar response in samples from 50 °C (30 min) when compared to the control (23 °C). In its turn, regarding the expression patterns of *PYRUVATE DEHYDROGENASE E1 COMPONENT SUBUNIT BETA-4*, *TRIFUNCTIONAL UDP-GLUCOSE A*,*6-DEHYDRATASE/UDP-4-KETO-6DEOXY-D-GLUCOSE 3*,*5-EPIMERASE/UDP-4-KETO-L-RHAMNOSE-REDUCTASE RHM1 (RHM1)*, *NADPH-CYTOCHROME P450 REDUCTASE 1 (ATR1)* and *PROBABLE FRUCTOKINASE-7*, samples from high temperatures generally presented lower expression relative to the control samples.

### 2.3. Quantification Analysis of Sugars and Sugars Alcohols

All sugars (sucrose, glucose, fructose, raffinose) and sugar alcohols (mannitol and sorbitol) analyzed were detected at EMs from different induction treatments. Nonetheless, raffinose was only detected in half of the analyzed samples, and no statistical analysis was performed regarding this data.

Significant differences between treatments were only found for sucrose concentrations (Appendix A). Samples from the control (23 °C) presented a significantly higher concentration of sucrose when compared to samples from 50 °C (30 min), with the other two treatments presenting an intermediate response (Table 2). The same pattern was found for glucose, the sugar that presented the highest concentration levels, despite the fact that no significant differences were found. Respecting the concentrations of fructose, mannitol and sorbitol, each presented different individual responses without substantial differences (Table 2).

### 2.4. Quantification Analysis of Amino Acids

Except for hydroxyproline, all other amino acids under study were detected in EMs from different induction treatments. The amino acids that presented higher concentrations were alanine and asparagine, but no significant differences were detected. In fact, statistically, significant differences between treatments were only found for the concentrations of glutamine, glycine and cysteine (Appendix A). Glutamine presented a significantly higher concentration in samples from 40 °C (4 h) compared to samples from the control (23 °C) and 50 °C (30 min). In turn, glycine and cysteine presented a significantly higher concentration at samples from the control (23 °C) when compared to samples from 50 °C (30 min) (Table 3). As previously observed for the concentrations of sugars, different amino acids presented individual patterns as result of the induction treatments applied (Table 3).

## 3. Discussion

In this report, in order to extend our knowledge of the stress mechanisms triggered by heat stress during the induction of SE of *P. halepensis*, we studied its effect on the proteome and the relative concentration of soluble sugars, sugar alcohols and amino acids.

For protein identification, a short gel liquid-chromatography coupled to tandem mass spectrometry (Short-GeLCMS/MS) approach, using DDA and DIA acquisition methods, was performed in EMs induced under different temperature treatments. Notwithstanding that the genome of *P. halepensis* is yet to be fully sequenced and may have negatively influenced the identification numbers, we were able to identify 1315 proteins and determine the relative concentration for 858. This is in agreement with the average number found in other proteomic studies in *Pinus* species [31,37,38,39].

The PLS-DA multivariate analysis permitted the selection of the 267 most significant proteins regarding the heat stress response, and the GO enrichment analysis disclosed that proteins involved in cellular metabolism and signaling were the most abundant. Accordingly, the 27 proteins identified with higher interest concerning heat stress response presented the same molecular functions. As reviewed by Pinheiro et al. [40], several proteome studies developed in Mediterranean woody species for the identification of stress-responsive proteins and tolerance/adaptation mechanisms under stress revealed high metabolic adjustment. Furthermore, in agreement with the data presented here, the rearrangement of various basal and secondary metabolic pathways such as glycolysis, the tricarboxylic acid cycle and secondary metabolites production has been found in different pine species as a general stress response [20,31,37,39,41,42,43].

Transcriptomics and proteomics have been used to identify heat-stress-responsive genes and proteins, including signaling components such as protein kinases and transcription factors, and functional genes, such as heat shock proteins (HSPs) and catalase [44]. In control temperature conditions, HSPs bind to heat shock factors, maintaining them in an inactive state. During stress, they are released and facilitate the repairing or removing of damaged proteins. The consequential release of the heat shock factors leads to an activation boost to produce more HSPs to protect the cells and restore the cellular balance between these components, diminishing the stress response [45]. HSPs allow other proteins to preserve their stability under heat stress by assisting in protein folding and processing [46], and its relation with heat stress response has been widely studied [47].

Accordingly, the HSP chaperonin CPN60 was at higher amounts in samples under temperature stress. However, 20 kDa chaperonin, a HSP that functions as a co-chaperone along with CPN60, was down-accumulated in samples under heat stress conditions. This can be related to the fact that it has been proved that this co-chaperone, alone, has other functions, such as the upregulation of superoxide dismutase genes in *A. thaliana* chloroplasts [48] and negative regulation of the abscisic acid signaling in the same species [49]. In this regard, in a previous study conducted in *P. halepensis* concerning the effect of heat stress in the relative expression of stress-related genes [28], no significant differences could be found regarding *SUPEROXIDE DISMUTASE* [*Cu–Zn*] concentrations at EMs, but a down-regulation was found for the in vitro somatic plants under stressful conditions.

RNA-binding proteins interact with mRNA via RNA-binding domains affecting RNA stability and accessibility for translation, regulating both RNA and protein expression [50]. In this report, the concentration of polyadenylate-binding protein RBP47 was lower at samples under heat stress treatments. In accordance with our results, in *A. thaliana*, this protein was reported to be located at the nucleus under control conditions while, under stress conditions, it was relocated to cytoplasmic granules [51]. The authors suggest that, under stress conditions, the concentration of these molecules can diminish rapidly due to their aggregation to mRNA polysomes to maintain cellular homeostasis. Agglutinin amounts were also decreased under stressful conditions. *Ricinus communis* agglutinin is one of the most important lectins and has been broadly used as an instrument to study cell surfaces and to purify glycans [52]. Lectins are proteins with diverse molecular structures capable to recognize and bind specifically and reversibly to carbohydrate structures and have been proven to be involved in stress response [53,54].

NADP-dependent malic enzyme 3 is a protein expressed by the *NADP-ME3* gene. According to Wheeler et al. [55], there are four NADP-dependent malic enzyme genes (*NADP-ME1-4*) in the *A. thaliana* genome, and the different isoforms of this enzyme are responsible to catalyze the reversible oxidative decarboxylation of L-malate to pyruvate, CO_2_ and NADPH in the presence of a divalent cation. However, they lead to the creation of proteins with unique regulatory mechanisms, and *NADP-ME3* is inhibited by fumarate with no modification of the enzymatic activity in the presence of aspartate and succinate [56]. Concerning its role in stress response, it appears that NADP-dependent malic enzyme plays an essential role in different biotic and abiotic stresses. As reviewed by Chen et al. [57], and in accordance with our results, in tobacco, chloroplastic NADP-dependent malic enzyme increased after polyethylene glycol and drought treatments, whereas the transcription of cytosolic *NADP-ME* remained similar or decreased. In *A. thaliana*, *NADP-ME2* appears to be involved in plant basal defense following pathogen recognition through the production of ROS [58]. In its turn, alcohol dehydrogenase, a protein that presented the same pattern as the previous, is responsible for converting alcohols to aldehydes in plants and is important for NAD metabolism during anaerobic respiration [59]. A proteomic study developed in *A. thaliana* ecotypes that developed contrasting efficient methods of adaptations to phosphate deficiency revealed that aconitate hydratase 2, alcohol dehydrogenase and malic enzyme consistently presented a reverse response between the two ecotypes [60]. Grapevine leaves overexpressing alcohol dehydrogenase suffered a drastic reduction of 90% of sucrose levels [61]. This contrasts with our results since the concentration of sucrose was significantly higher in samples from the control, the same treatment where we found a higher concentration of alcohol dehydrogenase.

Proteins involved in DNA binding and cellular division, namely dynamin-related protein 12A and actin-depolymerizing factor 10, presented higher amounts at EMs from heat stress conditions. Actin depolymerization has been shown to induce programmed cell death [62]. The application of high doses of an actin-depolymerizing drug at the proliferation stage of Norway spruce SE induced cell death of suspensor cells followed by disintegration of the meristematic centers and their subsequent death, while at the maturation stage, low doses led to suspensors cell death which accelerated and synchronized the development of high-quality embryos [63]. In this sense, it appears that the reorganization of cytoskeletal structures has an important role in programmed cell death as well as in the embryogenesis process.

The histone H4 variant TH091 and histone H2A.2.2 (core components of nucleosomes), and the 40S ribosomal protein S15a-1 and 60S ribosomal protein L18-3 (structural components of ribosomes) had increased amounts in samples from heat stress conditions. This agrees with previous studies on the temperature stress response, where ribosomal proteins, translation regulator factors and translation regulatory proteins played a significant role [39,64,65,66]. This data, combined with the fact that proteins´ abundance variations are not always correlated with changes in the corresponding transcriptome [67], can explain the lack of significant differences found in the transcription assay performed here. The protein-coding genes are transcribed to pre-messenger ribonucleic acid, further processed to messenger ribonucleic acid and finally translated into proteins, which in turn can be further processed and modified post-translationally [68]. Most differences found in the proteomic profiles may result from post-transcriptional regulation rather than transcription.

A parallel analysis of the proteome of EMs induced under heat stress (40 °C for 4 h and 60 °C for 5 min) of radiata pine was developed by our group [39]. The results found from the GO analysis showed the same over-representation of proteins regarding their biological processes (response to cadmium ion, proteasome mediated ubiquitin-dependent protein catabolic process, the glycolytic process and the tricarboxylic acid cycle); likewise, a deep reorganization of the machinery involved in protein synthesis, folding, transport and degradation was found. Notwithstanding, despite the fact that the proteins found as top-significant being involved in the same processes, the results regarding the effect of heat stress on their accumulation generally contrast; the few HSPs and chaperones, a variety of proteins related to oxidative stress response, and most of the enzymes directly involved in central metabolism (the glycolytic process and the tricarboxylic acid cycle) were present at significantly lower amounts in samples from the higher temperatures in *P. radiata*. Additionally, in somatic embryos produced under the same conditions in that species [31], the same pattern was found for central metabolism enzymes and proteins related to oxidative stress, while HSPs and chaperones had a contrary response. The hypothesis postulated by the results in *P. radiata* that HSPs and chaperones are important during long-term heat stress response, while proteins and metabolites involved in oxidative stress defense are required during earlier response stages, do not apply in *P. halepensis*.

As reviewed by Lamelas et al. [69], a great variety of metabolites of low molecular mass can avoid the damaging change in cellular components and restore homeostasis under stress. These include soluble carbohydrates such as glucose and fructose, amino acids and a variety of sugar alcohols. In this sense, regarding the quantification analysis of sugars, we found a significantly higher concentration of sucrose in control samples than those from the 50 °C (30 min) treatment. In this line, a study developed in *P. halepensis* seeds showed different concentrations of glucose, fructose and sucrose under drought stress, with the latest showing an enhanced accumulation in drought-tolerant seeds [70]. Furthermore, we found significant differences in the concentrations of glutamine, glycine, and cysteine between samples from different induction treatments. Glycine betaine seems to be important for abiotic stress tolerance; genes associated with glycine betaine synthesis have been transferred into plants such as maize, which do not accumulate glycine betaine, and enhance the level of synthesis upon stress [71]. In addition, in maize, the exogenous application of glycine and L-arginine to crops in order to analyze their effects on temperature stress showed that, while L-arginine led to more vigorous plants under stress, glycine presented none or negative effects [72]. Furthermore, in EMs of *P. radiata*, heat stress affected the abundance of phenolic compounds and several amino acids; tyrosine and isoleucine presented significantly lower concentrations in samples from the control, while leucine and histidine presented significant differences between the higher concentration at 50 °C (30 min) and the lower at 40 °C (4 h) [42]. Furthermore, while no effect was found in the concentration of soluble sugars in *P radiata* somatic embryos, sucrose levels were on the verge of statistical significance and decreased at higher temperatures [39]. In *P. radiata* Ems, significantly higher concentrations of fructose and glucose were found in samples from 40 °C (4 h), compared to the control and 60 °C (5 min). Altogether, it is clear that all these metabolites are highly involved in the heat stress response pathway, but it seems that the clear mechanisms are not fully defined yet.

## 4. Materials and Methods

### 4.1. SE Temperature Experiment and Plant Material Collection

SE induction of *Pinus halepensis* Ems under different temperatures was performed as described in [27]. One-year-old green female cones gathered from Manzanos (Spain; latitude: 42°44′29″ N, longitude: 2°52′35″ W, and enclosing immature seeds of *P. halepensis* from five open-pollinated trees were stored and prepared according to [18]. Whole megagametophytes containing immature embryos were used as initial explants and placed horizontally on a modified DCR [73] initiation medium supplemented with 30 g L^−1^ sucrose, 3.5 g L^−1^ gellan gum (Gelrite^®^, Duchefa Biochemie, Amsterdam, The Netherlands), a combination of 9.0 µM 2,4-dichlorophenoxyacetic acid and 2.7 µM kinetin and an EDM amino acid mixture [74]. For temperature treatments, Petri dishes containing initiation medium were preheated for 30 min, and immature megagametophytes were cultured at 40, 50 and 60 °C for 4 h, 30 min and 5 min, respectively. As control conditions, 23 °C was used and, after the application of the different treatments, all explants were kept at 23 °C in darkness.

After nine weeks on the initiation medium, initiation percentages were recorded and proliferating EMs were detached from the megagametophyte and transferred to the proliferation medium. This medium had the same composition as that used in the initiation stage, but a higher gellan gum concentration (4.5 g L^−1^). EMs were subcultured every two weeks and kept in the dark.

Following 4 proliferation subcultures, fresh tissue from five embryogenic cell lines (ECLs) per treatment were immersed in liquid nitrogen and immediately stored at −80 °C until further analysis.

### 4.2. Metabolites, RNA and Protein Extraction

Metabolites, RNA and protein extraction from five proliferating EMs (previously selected for maturation) per SE induction treatment was performed following a modified protocol described by [75].

#### 4.2.1. Metabolites Extraction

Two milliliters of cold (4 °C) metabolite extraction buffer (methanol:chloroform:water 2.5:1:0.5) was added to 500 mg FW of proliferation tissue (liquid nitrogen grinded) and mixed on a vortex. Samples were centrifuged (4700× *g*, 10 min, 4 °C) and the supernatant, containing metabolites, was transferred to 15 mL falcon tubes that contained 2 mL of phase separation mix (chloroform:water 1:1). Tubes were then centrifuged (4700× *g*, 10 min at room temperature) and two phases were clearly defined with a sharp interface. The upper and lower layers of polar and nonpolar metabolites were transferred to new falcons. These two layers were rewashed with 600 µL of phase separation buffer for a second fractionation and the upper layers, containing polar metabolites, were saved to new tubes and stored at −80 °C for further HPLC analysis.

#### 4.2.2. RNA and Proteins Extraction

Pellets containing proteins and nucleic acids were washed immediately with 2 mL of 0.75% (*v*/*v*) β-mercaptoethanol in 100% methanol, centrifuged (4700× *g*, 10 min, 4 °C) (×2) and air-dried. Pellets were dissolved in 1 mL of pellet solubilization buffer (PSB) (7 M guanidine HCl, 2% (*v*/*v*) Tween-20, 4% (*v*/*v*) Triton X - 100, 50 mM Tris, pH 7.5, 1% (*v*/*v*) β-mercaptoethanol) and incubated at 37 °C for 30 min. Samples were then centrifuged (4700× *g*, 6 min) and supernatants were transferred to new silica columns to bind DNA (Zymo Research, Irvin, CA, USA). After 1 min of incubation, columns were centrifuged (10,000× *g*, 1 min), and the flowthrough that contained RNA and proteins was immediately mixed with 600 µL of acetonitrile for total RNA extraction. The mix was transferred to a new silica column, incubated for 1 min and centrifuged (10,000× *g*, 1 min). The flowthrough was transferred to a new falcon tube and kept at 4 °C until protein isolation. Columns with bound RNA were washed with 750 µL of WB1 (2 mM Tris pH 7.5, 20 mM NaCl, 0.1 mM EDTA, 90% ethanol) and centrifuged (12,000× *g*, 2 min), and 50 µL of DNase I was added to each column and incubated for 15 min at room temperature. After nuclease treatment, the WB1 step was repeated, and then columns were washed with 750 µL of WB2 (2 mM Tris pH 7.5, 20 mM NaCl, 70% ethanol) and centrifuged (12,000× *g*, 2 min). Flowthrough was discarded, and columns were centrifuged again (14,000× *g*, 1 min) to dry the membrane completely. RNA was eluted from the column in 50 µL of RNase-free water. RNA samples were kept at −80 °C until further analysis.

Proteins were purified from the flowthrough using phenolic extraction: 1100 µL phenol (Tris-buffered, pH 8) and 1200 µL of water were added to each falcon. Samples were mixed on a vortex for 2 min and then centrifuged (4700× *g*, 8 min, room temperature). The supernatant, containing the phenolic phase, was transferred to a new falcon that contained 1200 µL of PWB (0.7 M sucrose, 50 mM Tris–HCl pH 7.5, 50 mM EDTA, 0.5% β-mercaptoethanol, 0.5% (*v*/*v*) Plant Protease Inhibitor Cocktail) and mixed thoroughly by vortex. After centrifugation (4700× *g*, 8 min, room temperature), upper phenolic phase was transferred to a new falcon, and proteins were precipitated by adding 3 mL of 0.1 M ammonium acetate and 0.5% β-mercaptoethanol in methanol in an overnight incubation at −20 °C. Samples were then centrifuged (4700× *g*, 20 min, 4 °C), and protein pellets were washed with acetone in an ultrasound bath until complete disaggregation of the pellet. Proteins were precipitated by centrifugation (4700× *g* for 20 min at 4 °C), and acetone was removed (2×). Protein pellets were allowed to air dry and resuspended in 80 µL of elution buffer (7M urea, 2M thiourea, 2% (*w*/*v*) CHAPS, 1% (*w*/*v*) DTT). Protein samples were stored at −20 °C until further analysis.

### 4.3. Protein Preparation and LC-MS Analysis

#### 4.3.1. Data-Dependent Acquisition (DIA) and Data-Independent Acquisition (DIA) Experiments

Protein extracts (70 µL) were precipitated with 4 volumes of cold acetone (280 µL) for 30 min at −80 °C. Samples were centrifuged (20,000× *g*, 4 °C, 20 min) and the pellet was resuspended in 50 µL of 1× Laemmli Sample Buffer. The total protein concentration was measured for each sample using the Pierce 660 nm Protein Assay kit (Thermo Scientific™, Waltham, MA, USA). For data-dependent acquisition (DDA) experiments, replicates from each condition were pooled into four different samples (Cond1: 23 °C, Cond2: 40 °C for 4h, Cond3: 50 °C for 30 min, and Cond4: 60 °C for 5 min) to create a library for each condition before sample processing, and for data-independent acquisition (DIA), each sample was processed individually for quantification purposes, as described by [76]. Protein content from each sample, adjusted based on the protein quantification values obtained previously, was separated by SDS-PAGE for about 17 min at 110 V (Short-GeLC Approach) and stained with Coomassie Brilliant Blue G-250. For DDA experiments, each lane was divided into 5 gel pieces in order to increase the depth of the analysis and to acquire more fragmentation spectra to correlate with the database and, for DIA experiments, they were divided into 3 gel pieces for further individual processing. After the destaining step with a 50 mM ammonium bicarbonate and 30% acetonitrile solution, gel bands were incubated overnight with trypsin for protein digestion, and peptides were extracted from the gel using 3 solutions containing different percentages of acetonitrile (30, 50, and 98%) with 1% formic acid. The organic solvent was evaporated using a vacuum-concentrator, and peptides were re-suspended in 30 µL of a solution containing 2% acetonitrile and 0.1% formic acid. Each sample was sonicated using a cup-horn (Ultrasonic processor, 750 W) for about 2 min, 40% amplitude and pulses of 1 sec ON/OFF. Ten microliters of each sample were analyzed by LC-MS/MS, either for DIA or DDA experiments.

#### 4.3.2. Protein LC-MS Analysis

Samples were analyzed on a NanoLC™ 425 System (Eksigent Framingham, MA, USA) coupled to a Triple TOF™ 6600 mass spectrometer (Sciex, Framingham, MA, USA). The ionization source was the OptiFlow^®^ Turbo V Ion Source equipped with the SteadySpray™ Low Micro Electrode (1–10 µL). The chromatographic separation was performed on a Triart C18 Capillary Column 1/32” (12 nm, S-3 µm, 150 × 0.3 mm, YMC) and using a Triart C18 Capillary Guard Column (0.5 × 5 mm, 3 μm, 12 nm, YMC) at 50 °C. The flow rate was set to 5 µL min^−1^ and mobile phases A and B were 5% DMSO plus 0.1% formic acid in water and 5% DMSO plus 0.1% formic acid in acetonitrile, respectively. The LC program was performed as follows: 5–35% of B (0–40 min), 35–90% of B (40–41 min), 90% of B (41–45 min), 90–5% of B (45–46 min) and 5% of B (46–50 min). The ionization source was operated in the positive mode set to an ion spray voltage of 4500 V, 10 psi for nebulizer gas 1 (GS1), 15 psi for nebulizer gas 2 (GS2), 25 psi for the curtain gas (CUR) and source temperature (TEM) at 100 °C. For DDA experiments, the mass spectrometer was set to scanning full spectra (*m*/*z* 350–1250) for 250 ms, followed by up to 100 MS/MS scans (*m*/*z* 100–1500). Candidate ions with a charge state between +1 and +5 and counts above a minimum threshold of 10 counts per second were isolated for fragmentation, and one MS/MS spectrum was collected before adding those ions to the exclusion list for 15 s (mass spectrometer operated by Analyst^®^ TF 1.7, Sciex^®^). The rolling collision was used with a collision energy spread of 5. For SWATH experiments, the mass spectrometer was operated in a looped product ion mode and specifically tuned to a set of 90 overlapping windows, covering the precursor mass range of 350–1250 *m*/*z*. A 50 ms survey scan (350–1250 *m*/*z*) was acquired at the beginning of each cycle, and SWATH-MS/MS spectra were collected from 100–1800 *m*/*z* for 35 ms resulting in a cycle time of 3.2 s.

### 4.4. Relative Expression of Stress-Related Transcripts

RNA quality and concentration were assessed using a spectrophotometer (NanoDrop™, Thermo Scientific, Waltham, MA, USA) and samples with a good quantity and integrity criteria were used for cDNA synthesis. An equal RNA quantity (1000 ng/sample) was reverse transcribed into cDNA, using NZY First-Strand cDNA Synthesis kit according to the instructions provided by the manufacturer. Three biological replicates were obtained for each treatment, e.g., cycle of temperature. Primer pairs were designed for transcripts of nine proteins previously identified as differentially accumulated between treatments, using the available NCBI Primer Design Tool, (amplicon length < 200 bp), and its specificity was validated through the amplification of a PCR product of the predicted size, using NZYTaq II 2× Green Master Mix kit, and visualization in a 1.5% (w/v) agarose gel electrophoresis. The melting curves evaluation corroborated the primer specificity as well.

Quantitative reverse transcription PCR (RT-qPCR) amplifications were performed in 96-well plates and in a CFX Connect^TM^ Real-Time System (Bio-Rad, Kaki Bukit, Republic of Singapore). PCR conditions were conducted under an initial denaturation at 95 °C for 2 min and 40 cycles of 5 s at 95 °C and 25 s at 55 °C. All the reactions were evaluated with two technical replicates and non-template controls were incorporated as well. Analyzed transcripts as well as primer pairs details are summarized in Table 4. The relative transcript levels were normalized using *ACTIN (ACT)* and *α-TUBULIN*, and the relative expression of each gene (R) was calculated based on ΔCt values using the following formula: R = 2^−ΔCt^ [77]. Finally, the fold changes between expression values obtained at the control treatment (23 °C) and different temperature treatments were calculated on logarithmic scale.

### 4.5. Quantification of Sugars and Sugars Alcohols

For sugars (sucrose, glucose, fructose and raffinose) and sugar alcohol (mannitol and sorbitol) quantification, 500 µL of the obtained polar metabolites was totally dried on a Speedvac, and pellets were re-suspended in 100 µL ultrapure water. The sugars were analyzed by HPLC using an Agilent 1260 Infinity II coupled to a refractive index detector (RID) (Agilent Technologies, Santa Clara, CA, USA). A Hi-Plex Ca column (7.7 × 300 mm, 8 µm) was used for sugar and sugar alcohol separation. The mobile phase was ultrapure water, and the samples were injected into the column at a flow rate of 0.2 mL min^−1^ at 80 °C for 40 min. Sugar concentrations were determined from internal calibration curves constructed with the corresponding commercial standards. Concentrations obtained from the HPLC analysis (g L^−1^) were conveniently adjusted, considering the initial concentration step (5 times), to express the results as µmol g FW^−1^.

### 4.6. Quantification of Amino Acids

For amino acids quantification, 500 µL of the obtained polar metabolites was totally dried on a Speedvac, and pellets were resuspended in 250 µL ultrapure water. Amino acid analysis was performed by HPLC using an Agilent 1260 Infinity II coupled to a fluorescence detector (FLD) (Agilent Technologies, Santa Clara, CA, USA). An AdvancedBio Amino Acid Analysis (AAA) column (100 mm, 2.7 µm Superficially Porous Particle (SPP)) was used for separation of 21 amino acids. The mobile phase A was 10mM Na_2_HPO_4_ + 10mM Na_2_B_4_O_7_, pH 8.2; and the mobile phase B was acetonitrile:methanol: water (45:45:10). The gradient program was the following: min 0–13.40, solvent A 98% and solvent B 2%; min 13.40–13.50, solvent A 43% and solvent B 57%; min 13.50–15.80, solvent B 100%; and min 15.80–18, solvent A 98% and solvent B 2%. The primary amino acids aspartic acid, glutamic acid, asparagine, serine, glutamine, histidine, glycine, threonine, arginine, alanine, tyrosine, cysteine, valine, methionine, tryptophan, phenylalanine, isoleucine, leucine, and lysine were derivatized with OPA and monitored at 338 nm. Hydroxyproline and proline were derivatized with FMOC and monitored at 262 nm. One µL of the sample was injected in the column at a flow rate of 1.5 mL min^−1^ for 18 min. Amino acid concentrations were determined from internal calibration curves constructed with the corresponding commercial standards. Concentrations obtained from the HPLC analysis (pmol µL^−1^) were conveniently adjusted, considering the initial concentration step (2 times), to express the results as µmol g FW^−1^.

### 4.7. Data Analysis

#### 4.7.1. Ion-Library Construction (DDA Information)

A specific ion-library of the precursor masses and fragment ions was created by combining all files from the DDA experiments in one protein identification search using the ProteinPilot™ software (v5.0, Sciex^®^). The paragon method parameters were the following: searched against the reviewed Viridiplantae database (Swissprot) downloaded on 1^st^ April from UniProtKB (The UniProt Consortium, 2019), cysteine alkylation by acrylamide, digestion by trypsin, and gel-based ID. An independent False Discovery Rate (FDR) analysis, using the target-decoy approach provided by Protein Pilot™, was used to assess the quality of identifications.

#### 4.7.2. Relative Quantification of Proteins (SWATH-MS)

SWATH data processing was performed using the SWATH^TM^ processing plug-in for PeakView^TM^ (v2.0.01, Sciex^®^). Protein relative quantification was performed in all samples using the information from the protein identification search. Quantification results were obtained for peptides with less than 1% of FDR and by the sum of up to 5 fragments/peptides. Each peptide was normalized for the total sum of areas for the respective sample. Protein relative quantities were obtained by the sum of the normalized values for up to 15 peptides/protein. A correlation analysis between samples was performed using the Spearman Rank Correlation method and considering the relative quantification values determined for all the samples to ensure that those from the same condition showed the same behavior.

#### 4.7.3. Statistical Analysis

A one-way analysis of variance was carried out to assess the effect of different induction treatments on concentrations of sugars, sugar alcohols, stress-related transcripts expression and amino acids. An ANOVA was made after confirmation of the homogeneity of variances and normality of the samples. In the cases that data did not fulfill the homogeneity of variances and normality of the samples, the corresponding non-parametric test, Kruskal–Wallis test, was applied. When significant differences were found (*p* < 0.05), the Tukey HSD post hoc test or Dunn’s multiple comparison test, respectively, were conducted to determine which treatments were statistically different.

Two different approaches were employed to analyze the protein results, combining multivariate and univariate analyses. First, a partial least square-discriminant analysis (PLS-DA) using the MetaboAnalyst web-based platform [78] was performed to find out the separation between the four conditions and simultaneously identify the most significant top protein features able to classify the four groups based on variable influence on projections (VIP) values. Those proteins were then clustered based on their biological function according to FunRich and the Plants database from the UniProt database. A correlation analysis between samples was performed, showing that sample #04 is less correlated with the others in the same group. Nonetheless, the statistical analysis was performed including that sample. For the univariate analysis, as cross-validation, a Kruskal–Wallis test was performed to select the proteins which were statistically different between the 4 conditions. Dunn’s test of multiple comparisons, with Benjamini-Hochberg p-value adjustment, was performed to determine which statistical differences were observed. Finally, the 27 proteins overlapping on both approaches were selected, and cluster analysis was performed. The MetaboAnalyst web-based platform was used in order to investigate their relation and relative abundance by generating heatmap and correlation matrix plots. The Euclidean distance and the Complete algorithm were used for the heatmap hierarchical clustering, whereas for the correlation matrix the Pearson’s correlation test was applied.

## 5. Conclusions

As far as it is known, this is the first report studying the effect of heat stress on the proteome and the concentration of soluble sugars, sugar alcohols and amino acids of *P. halepensis* EMs.

Heat stress during the induction phase of SE led to proteomic and metabolomic reorganization; several enzymes directly involved in metabolic pathways as well as proteins previously found to be involved in temperature stress response such as histones (histone H4 variant TH091 and histone H2A.2.2) and ribosomal proteins (40S ribosomal protein S15a-1 and 60S ribosomal protein L18-3) presented higher amounts at higher temperatures. In addition, sucrose, glycine and cysteine presented lower concentrations at higher temperatures, while the concentrations of glutamine were higher, emphasizing the direct involvement of these primary metabolites in stress response.

The results of this work reinforce the idea that a temperature priming effect during the initial stages of somatic embryogenesis can trigger long-lasting effects in the mechanisms involved in heat stress response.

## Figures and Tables

**Figure 1 ijms-24-07211-f001:**
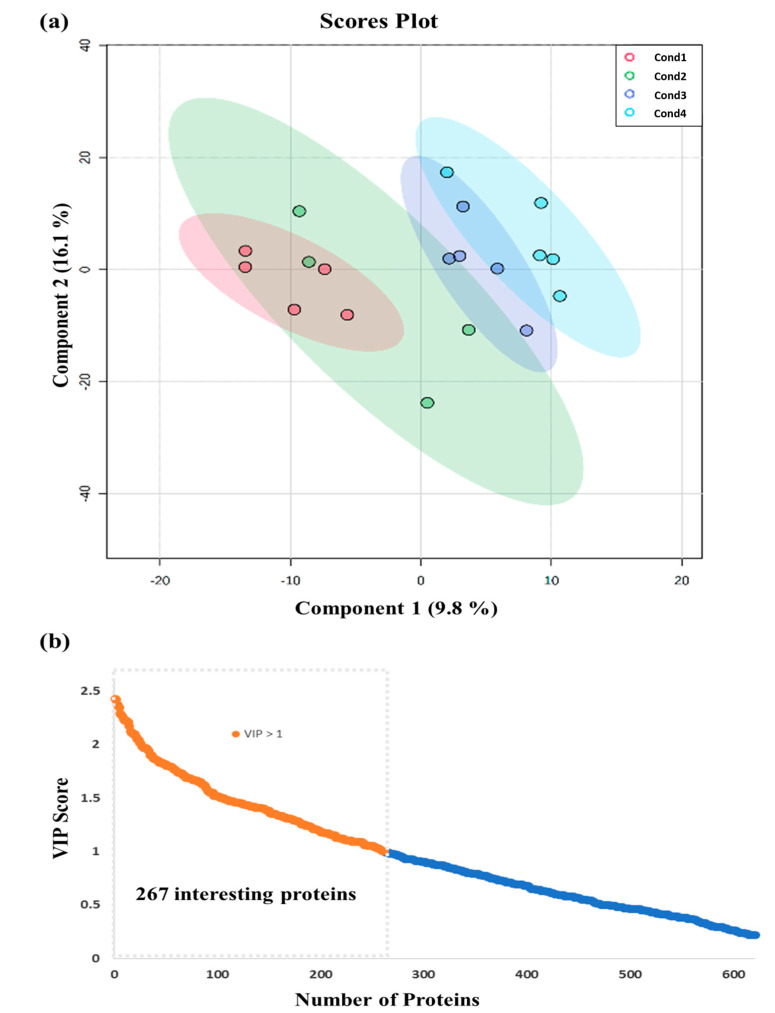
PLS-DA multivariate analysis of the detected 858 proteins from embryonal masses induced under four different temperature treatments (Cond1: 23 °C; Cond2: 40 °C (4 h); Cond3: 50 °C (30 min); Cond4: 60 °C (5 min)): (**a**) bi-dimensional representation of the scores. Data normalization was performed using the AutoScale method, and the scores of the two first components are represented, showing the ovals at 95% confidence interval; (**b**) variable importance in projection (VIP) scores. From the 858 quantified proteins, 267 were selected as interesting variables for the group separation observed in the scores plot. A cut-off of 1 was considered to select the important variables.

**Figure 2 ijms-24-07211-f002:**
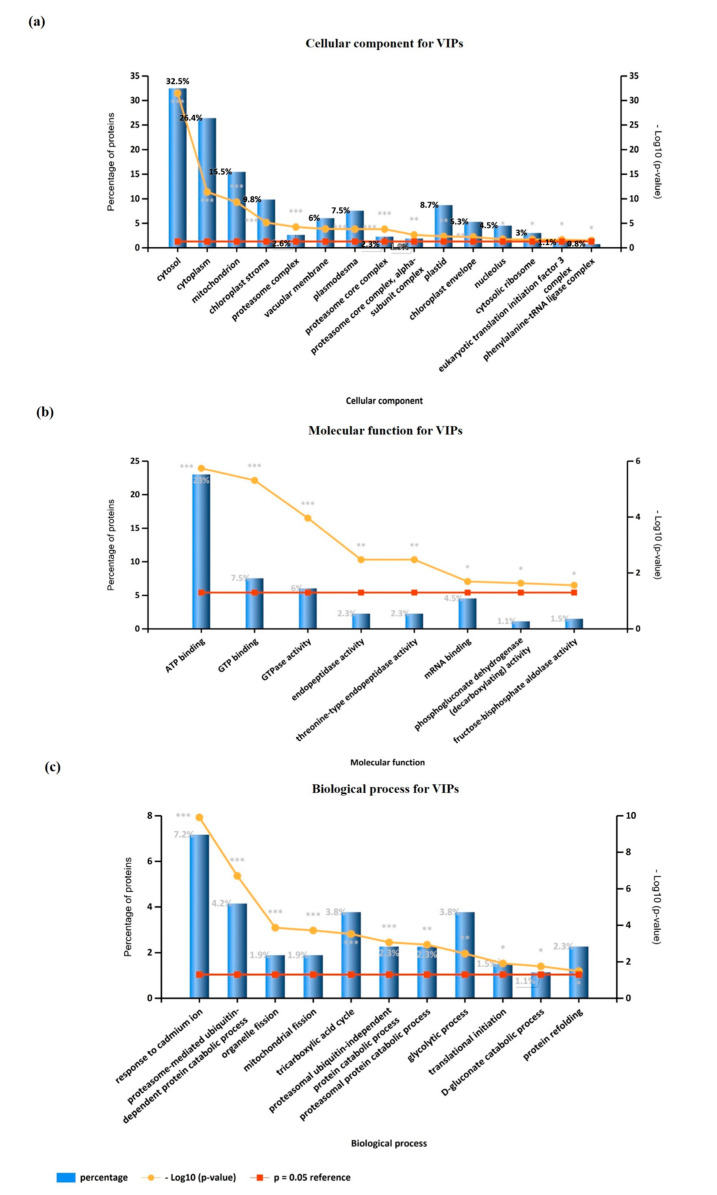
Gene ontology enrichment analysis of the 267 VIP proteins, selected by PLS-DA analysis, performed using FunRich and the Plants database from UniProt database: (**a**) cellular component; (**b**) molecular function, and (**c**) biological process. *: *p* < 0.05; **: *p* < 0.01 ***: *p* < 0.001.

**Figure 3 ijms-24-07211-f003:**
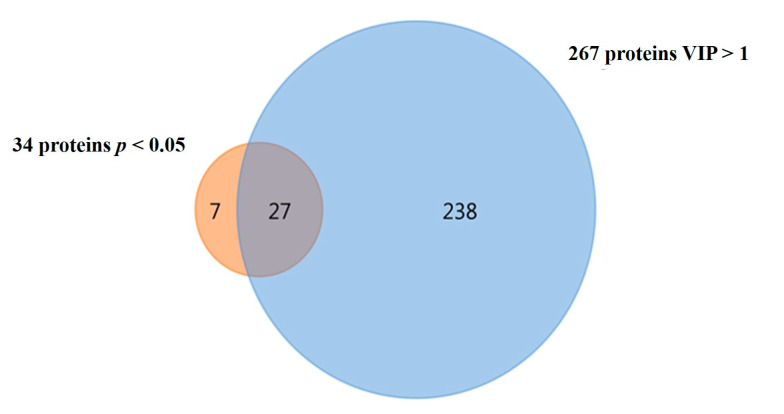
Venn diagram showing that 27 proteins were common between the significant proteins from the univariate analysis (Kruskal–Wallis, *p* < 0.05) and the multivariate analysis (PLS-DA, VIP > 1).

**Figure 4 ijms-24-07211-f004:**
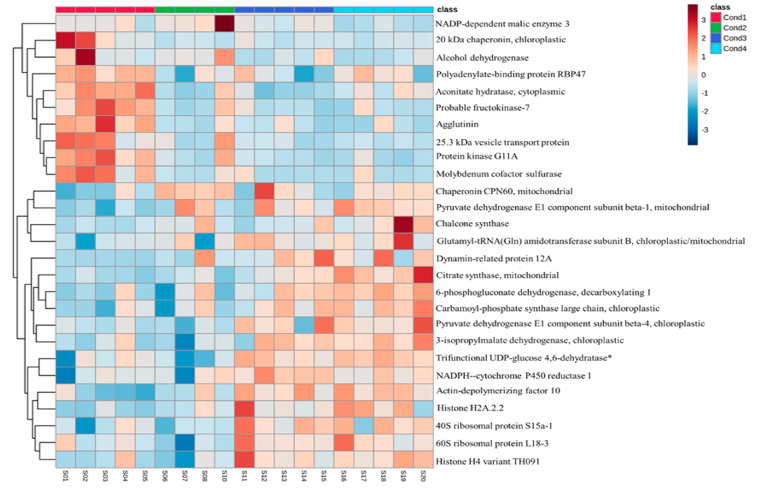
Hierarchical clustering heatmap using the 27 proteins selected, with the combination of univariate and multivariate analysis, from EMs induced under four different temperature treatments (Cond1: 23 °C; Cond2: 40 °C (4 h); Cond3: 50 °C (30 min); Cond4: 60 °C 5 (min)); S01–S20, sample1 to sample20. Hierarchical clustering was performed only at the protein (rows) level using Euclidean distance and complete for the clustering algorithm. * Trifunctional UDP-glucose 4.6-dehydratase/UDP-4-keto-6-deoxy-D-glucose 3.5-epimerase/UDP-4-keto-L-rhamnose-reductase RHM1.

**Figure 5 ijms-24-07211-f005:**
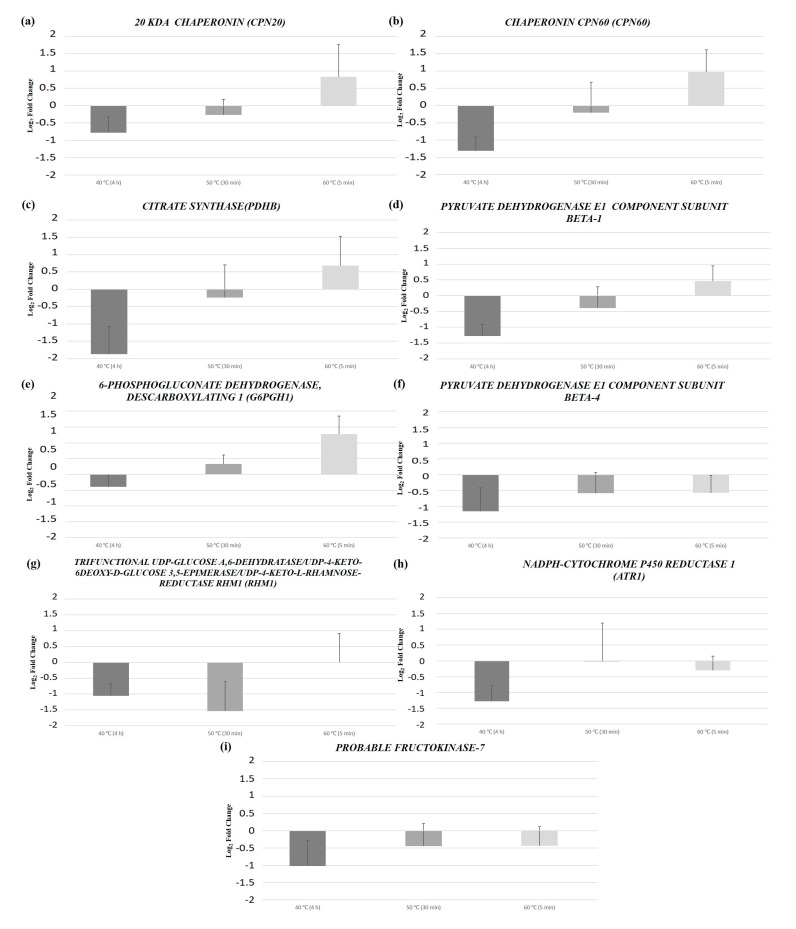
Relative expression fold change, with respect to the control (23 °C), of the nine transcripts from *P. halepensis* EMs induced under different temperature treatments [40 °C (4 h), 50 °C (30 min) and 60 °C (5 min)]: (**a**) *20 kDa CHAPERONIN (CPN20)*; (**b**) *CHAPERONIN CPN60* (*CPN60)*; (**c**) *CITRATE SYNTHASE (PDHB)*; (**d**) *PYRUVATE DEHYDROGENASE E1 COMPONENT SUBUNIT BETA-1*; (**e**) *6-PHOSPHOGLUCONATE DEHYDROGENASE*, *DECARBOXYLATING 1 (G6PGH1)*; (**f**) *PYRUVATE DEHYDROGENASE E1 COMPONENT SUBUNIT BETA-4*; (**g**) *TRIFUNCTIONAL UDP-GLUCOSE A*,*6-DEHYDRATASE/UDP-4-KETO-6DEOXY-D-GLUCOSE 3*,*5-EPIMERASE/UDP-4-KETO-L- RHAMNOSE-REDUCTASE RHM1 (RHM1)*; (**h**) *NADPH-CYTOCHROME P450 REDUCTASE 1 (ATR1)*; (**i**) *PROBABLE FRUCTOKINASE-7*. Data are presented as mean values + SE.

**Table 1 ijms-24-07211-t001:** Details of the twenty-seven proteins with higher interest, selected from the combination of the multivariate (PLS-DA) and the univariate statistical analysis (Kruskal–Wallis), detected in *P. halepensis* EMs induced under different temperature treatments: Cond1: 23 °C; Cond2: 40 °C (4 h); Cond3: 50 °C (30 min); Cond4: 60 °C 5 (min).

			Fold Change			Kruskal–Wallis	PLS-DA
Protein	Uniprot Accession	Species	Cond2/Cond1	Cond3/Cond1	Cond4/Cond1	*p*-Value	VIP
NADP-dependent malic enzyme 3	Q9XGZ0	*Arabidopsis thaliana*	2.25	0.96	0.44	0.006	1.26
20 kDa chaperonin, chloroplastic	O65282	*Arabidopsis thaliana*	0.08	0.16	0.13	0.018	2.08
Alcohol dehydrogenase	P17648	*Fragaria ananassa*	0.79	0.34	0.21	0.041	2.00
Polyadenylate-binding protein RBP47	Q9LEB3	*Nicotiana plumbaginifolia*	0.47	0.49	0.67	0.015	1.45
Aconitate hydratase, cytoplasmic	P49608	*Cucurbita maxima*	0.39	0.26	0.51	0.006	2.30
Probable fructokinase-7	Q9FLH8	*Arabidopsis thaliana*	0.30	0.20	0.23	0.024	2.74
Agglutinin	P06750	*Ricinus communis*	0.23	0.22	0.18	0.011	2.80
25.3 kDa vesicle transport protein	Q94AU2	*Arabidopsis thaliana*	0.44	0.20	0.20	0.036	2.59
Protein kinase G11A	Q0DCT8	*Oryza sativa* subsp. *japonica*	0.38	0.17	0.15	0.014	3.05
Molybdenum cofactor sulfurase	Q8LGM7	*Solanum lycopersicum*	0.16	0.11	0.13	0.016	2.65
Chaperonin CPN60, mitochondrial	P35480	*Brassica napus*	2.84	2.13	2.01	0.025	1.08
Pyruvate dehydrogenase E1 component subunit beta-1, mitochondrial	Q6Z1G7	*Oryza sativa* subsp. *japonica*	1.21	1.18	1.28	0.034	2.47
Chalcone synthase	P30079	*Pinus sylvestris*	2.02	1.60	3.19	0.031	1.95
Glutamyl-tRNA(Gln) amidotransferase subunit B, chloroplastic/mitochondrial	Q2R2Z0	*Oryza sativa* subsp. *japonica*	1.18	1.64	1.79	0.036	2.14
Dynamin-related protein 12A	Q39821	*Glycine max*	2.55	3.92	3.58	0.023	2.02
Citrate synthase, mitochondrial	O80433	*Daucus carota*	1.12	1.55	2.39	0.0040	3.08
6-phosphogluconate dehydrogenase, decarboxylating 1	Q9LI00	*Oryza sativa* subsp. *japonica*	1.00	1.17	1.28	0.026	2.71
Carbamoyl-phosphate synthase large chain, chloroplastic	B9EXM2	*Oryza sativa* subsp. *japonica*	1.02	1.18	1.29	0.042	2.57
Pyruvate dehydrogenase E1 component subunit beta-4, chloroplastic	Q10G39	*Oryza sativa* subsp. *japonica*	0.80	1.48	1.63	0.025	2.13
3-isopropylmalate dehydrogenase, chloroplastic	P29696	*Solanum tuberosum*	0.82	1.32	1.46	0.038	2.35
Trifunctional UDP-glucose 4.6-dehydratase/UDP-4-keto-6-deoxy-D-glucose 3.5-epimerase/UDP-4-keto-L-rhamnose-reductase RHM1	Q9SYM5	*Arabidopsis thaliana*	0.68	1.49	1.60	0.009	2.39
NADPH--cytochrome P450 reductase 1	Q9SB48	*Arabidopsis thaliana*	1.07	1.58	1.34	0.019	1.79
Actin-depolymerizing factor 10	Q9LQ81	*Arabidopsis thaliana*	1.24	1.53	1.56	0.017	3.03
Histone H2A.2.2	P02277	*Triticum aestivum*	2.34	4.10	4.50	0.023	2.53
40S ribosomal protein S15a-1	P42798	*Arabidopsis thaliana*	0.59	1.26	1.26	0.018	1.62
60S ribosomal protein L18-3	Q940B0	*Arabidopsis thaliana*	0.96	1.35	1.25	0.028	2.08
Histone H4 variant TH091	P62786	*Triticum aestivum*	0.78	1.50	1.50	0.023	2.15

**Table 2 ijms-24-07211-t002:** Concentration of sugars (µmol g^−1^ FW) detected in *P. halepensis* EMs, collected at proliferation phase, induced under different temperatures.

Sugars (µmol g^−1^ FW)	Cond1 23 °C	Cond2 40 °C (4 h)	Cond3 50 °C (30 min)	Cond4 60 °C (5 min)
Sucrose	5.39 ± 0.75 ^a^	3.35 ± 0.94 ^ab^	2.26 ± 0.53 ^b^	2.77 ± 0.55 ^ab^
Glucose	40.90 ± 1.86 ^a^	40.81 ± 4.29 ^a^	32.24 ± 5.68 ^a^	35.28 ± 2.71 ^a^
Fructose	24.48 ± 1.11 ^a^	29.53 ± 2.60 ^a^	21.21 ± 4.52 ^a^	21.57 ± 1.69 ^a^
Mannitol	0.18 ± 0.06 ^a^	0.32 ± 0.10 ^a^	0.34 ± 0.06 ^a^	0.41 ± 0.14 ^a^
Sorbitol	0.14 ± 0.04 ^a^	0.16 ± 0.06 ^a^	0.13 ± 0.04 ^a^	0.13 ± 0.03 ^a^

Data are presented as mean values ± SE. Significant differences at *p* < 0.05 within a line are indicated by different letters.

**Table 3 ijms-24-07211-t003:** Concentration of amino acids (µmol g^−1^ FW) detected in *P. halepensis* EMs, collected at proliferation phase, induced under different temperatures.

Amino Acids (µmol g^−1^ FW)	Cond1 23 °C	Cond2 40 °C (4 h)	Cond3 50 °C (30 min)	Cond4 60 °C (5 min)
Aspartic acid	0.135 ± 0.004 ^a^	0.126 ± 0.019 ^a^	0.109 ± 0.026 ^a^	0.137 ± 0.019 ^a^
Glutamic acid	0.428 ± 0.011 ^a^	0.465 ± 0.061 ^a^	0.375 ± 0.073 ^a^	0.445 ± 0.043 ^a^
Asparagine	0.725 ± 0.111 ^a^	0.994 ± 0.038 ^a^	0.760 ± 0.116 ^a^	0.849 ± 0.094 ^a^
Serine	0.315 ± 0.023 ^a^	0.288 ± 0.029 ^a^	0.253 ± 0.035 ^a^	0.289 ± 0.027 ^a^
Glutamine	0.449 ± 0.032 ^b^	0.734 ± 0.102 ^a^	0.430 ± 0.009 ^b^	0.564 ± 0.076 ^ab^
Histidine	0.057 ± 0.006 ^a^	0.069 ± 0.015 ^a^	0.059 ± 0.009 ^a^	0.053 ± 0.005 ^a^
Glycine	0.331 ± 0.027 ^a^	0.256 ± 0.023 ^ab^	0.230 ± 0.020 ^b^	0.292 ± 0.019 ^ab^
Threonine	0.102 ± 0.009 ^a^	0.110 ± 0.012 ^a^	0.086 ± 0.012 ^a^	0.098 ± 0.011 ^a^
Arginine	0.266 ± 0.036 ^a^	0.349 ± 0.078 ^a^	0.353 ± 0.062 ^a^	0.356 ± 0.105 ^a^
Alanine	0.891 ± 0.029 ^a^	0.851 ± 0.002 ^a^	0.824 ± 0.028 ^a^	0.867 ± 0.011 ^a^
Tyrosine	0.055 ± 0.004 ^a^	0.063 ± 0.009 ^a^	0.052 ± 0.005 ^a^	0.056 ± 0.005 ^a^
Cysteine	0.369 ± 0.028 ^a^	0.277 ± 0.028 ^ab^	0.254 ± 0.023 ^b^	0.322 ± 0.017 ^ab^
Valine	0.159 ± 0.012 ^a^	0.157 ± 0.019 ^a^	0.136 ± 0.018 ^a^	0.154 ± 0.020 ^a^
Methionine	0.020 ± 0.001 ^a^	0.021 ± 0.001 ^a^	0.019 ± 0.001 ^a^	0.020 ± 0.001 ^a^
Tryptophan	0.068 ± 0.004 ^a^	0.068 ± 0.009 ^a^	0.061 ± 0.006 ^a^	0.061 ± 0.009 ^a^
Phenylalanine	0.071 ± 0.004 ^a^	0.068 ± 0.007 ^a^	0.071 ± 0.008 ^a^	0.068 ± 0.007 ^a^
Isoleucine	0.082 ± 0.006 ^a^	0.090 ± 0.012 ^a^	0.079 ± 0.012 ^a^	0.080 ± 0.011^a^
Leucine	0.063 ± 0.006 ^a^	0.072 ± 0.015 ^a^	0.067 ± 0.008 ^a^	0.061 ± 0.006 ^a^
Lysine	0.092 ± 0.006 ^a^	0.113 ± 0.025 ^a^	0.085 ± 0.013 ^a^	0.098 ± 0.010 ^a^
Proline	0.105 ± 0.018 ^a^	0.141 ± 0.036 ^a^	0.104 ± 0.032 ^a^	0.098 ± 0.032 ^a^

Data are presented as mean values ± SE. Significant differences at *p* < 0.05 within a line are indicated by different letters.

**Table 4 ijms-24-07211-t004:** List of primers used in quantitative reverse transcription PCR (RT-qPCR) for relative expression analysis. Name of transcript, forward and reverse primer sequences, amplification length and melting temperatures of primers are described.

Name	Forward (5′ → 3′)	Reverse (5′ → 3′)	Amplification Length	Tm (°C)
*ACTIN (ACT)*	CACTGCACTTGCTCCCAGTA	AACCTCCGATCCAAACACTG	130	56
*α-TUBULIN*	ATCTGGAGCCGATGTCA	TGATAAGCTGTTAGGATGGAA	75	55
*PYRUVATE DEHYDROGENASE E1 COMPONENT SUBUNIT BETA-4*	TGCGCATGTACCAGGATTGA	AACTTCCGCAGAAACAGGGA	151	57
*CHAPERONIN CPN60 (CPN60)*	CAAACAGGTTGCTAACCGCC	TTGCATTCATTCCAGCAGCG	128	57
*NADPH-CYTOCHROME P450 REDUCTASE 1 (ATR1)*	GAGCCTACTGACAATGCTGCC	GGCGATTACCAAGAGCAAACAC	109	58
*RHM1 **	TATCGCTAGTGCTGACTTGGT	CCGAAGGAATTGTCGACGTC	99	56
*PYRUVATE DEHYDROGEN-ASE E1 COMPONENT SUBU-NIT BETA-4*	CATAAGGAGCGAGAACCCCG	CGCGAGTATGTGAGGATGGT	150	57
*6-PHOSPHOGLUCONATE DEHYDROGENASE, DECARBOXYLATING 1 (G6PGH1)*	ATGGGAGTTTCGGGTGGAGA	AAGCAACACACGGTCCACTAT	142	58
*20 KDA CHAPERONIN (CPN20)*	AACAGCTGGAGGGTTGTTGTT	CCTTCCTCGTCTAGGGAACC	96	57
*PROBABLE FRUCTOKINASE-7*	CTGACCGGTGGTGATGATCC	TCTCCTGCACCGGTTGTATC	179	57
*CITRATE SYNTHASE (PDHB)*	TGGACATGGTGTTCTGCGTAA	ACCCCACTATGGGCATCAAC	186	57

* TRIFUNCTIONAL UDP-GLUCOSE A,6-DEHYDRATASE/UDP-4-KETO-6DEOXY-D-GLUCOSE 3,5-EPIMERASE/UDP-4-KETO-L-RHAMNOSE-REDUCTASE RHM1 (RHM1).

## Data Availability

Data is contained within the article or Appendix A.

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
