# Peer review of "Proteomic and Metabolic Analysis of Pinus halepensis Mill. Embryonal Masses Induced under Heat Stress"

_ijms, 2023, doi:10.3390/ijms24087211_

Round 1

Reviewer 1 Report

Pereira et al. studied the stress mechanisms triggered by temperature stress during the induction of SE in Aleppo pine. The study of the proteome of EMs produced under high temperatures was conducted.

The manuscript is generally well-written and structured. The analysis was successful, and the data was well understood and modeled in detail. In addition, the manuscript contains relevant paragraphs that have been discussed. The selection of the bibliography is appropriate to the content of the manuscript. I enjoyed reading the manuscript; congratulation to the authors.

However, some minor errors appeared throughout the manuscript.

-        Authors should scan the manuscript for minor punctuation and English errors (see attached file).

-        In material and methods, sections (4.2. Metabolites, RNA, and protein extraction and 4.3. Protein LC-MS analysis) is very long and must divide into two subsections.

Reviewer 2 Report

The current manuscript findings is very interesting and an important addition to our understanding of stress induced changes during the somatic embryogenesis in Pinus halepensis and its after effect on plant stress tolerance

standard methodology has been followed to perform the experiments.

The manuscript can be accepted in its current form. 

Reviewer 3 Report

The article (ijms-2274653) entitled “Heat Stress During Induction of Somatic Embryogenesis in Pinus halepensis– Proteomics and Metabolic Analysis”. In Aleppo pine, somatic embryogenesis was induced under different heat stress treatments and studied its effects on the proteome and the relative concentration of soluble sugars, sugar alcohols and amino acids. the authors have identified proteins related to the heat stress response. 

The same team has done a lot of similar research on Pinus halepensis & Pinus radiata, to name a few

1. Heat Stress in Pinus halepensis Somatic Embryogenesis Induction: Effect in DNA Methylation and Differential Expression of Stress-Related Genes Plants 2021, 10(11), 2333; https://doi.org/10.3390/plants10112333

2.Proteome-Wide Analysis of Heat-Stress in Pinus radiata Somatic Embryos Reveals a Combined Response of Sugar Metabolism and Translational Regulation Mechanisms Front. Plant Sci. 12:631239.doi: 10.3389/fpls.2021.631239

3.Thermopriming-associated proteome and sugar content responses in Pinus radiata embryogenic tissue Plant Sci. 2022, 321, 746 111327.

Since the work is similar to previous research, there is no novelty. I suggest that the authors do some comparative analysis with the Primus radiata and other species related to identified proteomics and Metabolites. Discuss in detail in the discussion about the common or novel proteins or metabolites involved in stress response mechanisms and how to apply them in forest management and breeding programs.

Reviewer 4 Report

The authors investigated the effects of heat stress on the proteome and metabolites of Aleppo pine embryonal masses. Some corrections are required for acceptance.

Abstract, L.20. The Latin name of the species should be given at the first mention of Aleppo pine.

Introduction.

There are more than a hundred species of pines and therefore a brief description of Aleppo pine needs to be provided.

L.61-63. It is better to transfer a detailed description of the heat treatments from the Introduction to other sections.

Results.

Subsection 2.1. Is this a new experiment or previously described [23]? If new, then it is incorrect to present the data of previous experiment, since they may differ.

Tables 2 and 3. Embryonal masses contain little dry matter and therefore the content of metabolites is more correctly represented by dry weight.

Table 4. Please use lower case letters for protein names. The same for Table S2 and in the text.

Fig. 5. The quality should be improved.

Discussion. Perhaps the similarities and differences between these results and similar work on Pinus radiata [35] should be described in detail.

Reviewer 5 Report

The manuscript, Heat Stress during Induction of Somatic Embryogenesis in Pinus halepensis – Proteomics and Metabolic Analysis, reports new information about the extended effects of the heat-stress treatment applied at SE induction stage to tissue proliferation stage during somatic embryogesis in Pinus halepensis. Its contribution could be publishable after revisions.

I have listed my major concerns/comments about this manuscript.

Comments:

This research project is interesting, which studies how the temperature treatments applied at the early somatic embryogenesis affect the following stages, i.e. tissue proliferation, embryo maturation, and etc.  The heat-stress treatments that were reported previously (ref. 23) and mentioned in this manuscript are more likely, to me, the explant (immature megagametophyta) treatments under higher temperatures, especially, the 60 C treatment for five minutes.  In order to avoid genotype effect and respond to specific temperature treatment quickly, embryonic mass might be the better experimental material for this kind of study in future.

Suggestions:

1)    I suggest the following title that fits better for this manuscript:

Proteomics and metabolic analysis of the embryonic tissues induced with heat-stress treatments in Pinus halepensis

2)    Results 2.1 should be deleted completely, since this content has been published previously (Ref. 23).  The related information/data could be mentioned in the sections of induction and/or discussion, but not in the section of results.

3)    As the mentioned in 2), the related information in the section of M+M should be moved to other sections.

4)    Line 416   Metabolites, RNA and protein extraction: five proliferating EMs per SE induction treatment was extracted respectively or using a mixture together?

5)    In the section of discussion, please enhance the significance of the current findings in understanding the long-lasting effects caused by the heat stress, such as the difference in SE maturation.  Please clarify whether the difference in embryo maturation resulted from different embryogenicity of the embryonic tissues (based on the number of total embryos per unit initial tissue) or the difference in the ability of embryo development.  The total number of embryos should include the visible embryos at various stages.

6)    Line 538   sugars alcohol   -  sugar alcohol 

Round 2

Reviewer 3 Report

As suggested, the authors added little information regarding the differences with P. radiata in the discussion.